# Culture Positivity and Antibiotic Resistance in Respiratory Intensive Care Patients: Evaluation of Readmission and Clinical Outcomes

**DOI:** 10.3390/diagnostics15141737

**Published:** 2025-07-08

**Authors:** Oral Menteş, Deniz Çelik, Murat Yildiz, Kerem Ensarioğlu, Maşide Ari, Mustafa Özgür Cırık, Abdullah Kahraman, Zehra Nur Şeşen, Savaş Gegin, Yusuf Taha Güllü

**Affiliations:** 1Department of Intensive Care, Gulhane Training and Research Hospital, 06010 Ankara, Turkey; omentes@live.com; 2Department of Pulmonology, Faculty of Medicine, Alanya Alaaddin Keykubat University, 07425 Antalya, Turkey; drdenizcelik@hotmail.com; 3Department of Pulmonology, Ankara Ataturk Sanatorium Training and Research Hospital, University of Health Sciences, 06200 Ankara, Turkey; kerem.ensarioglu@gmail.com; 4Department of Pulmonology, Faculty of Medicine, Atatürk Sanatorium Research Hospital, University of Health Sciences, 06200 Ankara, Turkey; masidetuten@icloud.com; 5Department of Anesthesiology, Ankara Ataturk Sanatorium Training and Research Hospital, University of Health Sciences, 06200 Ankara, Turkey; dr.ozgurr@hotmail.com; 6İntensive Care Unit, Etlik City Hospital, University of Health Sciences, 06200 Ankara, Turkey; abdullahhero100@gmail.com; 7Department of Infection Diseases, Ankara Ataturk Sanatorium Training and Research Hospital, University of Health Sciences, 06200 Ankara, Turkey; zsesen@hotmail.com; 8Department of Pulmonology, Samsun Training and Research Hospital, 55090 Samsun, Turkey; geginn@hotmail.com; 9Anesthesia Program Mudanya, Vocational School, Mudanya University, 16960 Bursa, Turkey; tahagullu@gmail.com

**Keywords:** multidrug-resistant bacteria (MDRB), respiratory intensive care unit, readmission rates, culture positivity, C-reactive protein (CRP), respiratory infections

## Abstract

**Background:** Multidrug-resistant bacteria (MDRB) represent a significant challenge in intensive care units (ICUs), as they limit treatment options, prolong hospital stays, and escalate healthcare costs. Respiratory ICUs are particularly affected due to the high prevalence of chronically ill patients with recurrent infections. Understanding the impact of culture positivity and MDRB on clinical outcomes and readmission rates is essential for enhancing patient care and addressing the growing burden of antimicrobial resistance. **Methods:** This retrospective study was conducted in a specialized respiratory ICU at a tertiary care hospital between 1 January 2019, and 1 January 2020. A total of 695 ICU admissions were analyzed, with patients grouped based on readmission status and culture results. Demographic, clinical, and laboratory data were reviewed. Statistical analyses were performed using appropriate tests, with *p*-values ≤ 0.05 considered statistically significant. **Results:** Among the 519 unique patients, 65 experienced ICU readmissions. Male patients were significantly more likely to be readmitted (*p* = 0.008). Culture positivity was predominantly observed in respiratory samples, with *Klebsiella* spp. identified as the most common pathogen. MDRB prevalence exceeded 60% in both groups, significantly prolonging ICU stays (*p* = 0.013). However, no significant differences in survival rates were observed between MDRB-positive and MDRB-negative groups. Notably, patients with readmissions had lower C-reactive protein (CRP) levels both during admission and at discharge compared to non-readmitted patients (*p* = 0.004). This paradox may reflect a subclinical inflammatory response associated with bacterial colonization rather than active infection, particularly in patients with chronic respiratory diseases. **Conclusions:** MDRB infections and culture positivity are key contributors to prolonged ICU stays, resulting in increased healthcare costs. Implementing effective strategies to manage MDRB infections is critical for improving outcomes in respiratory ICUs and reducing associated risks. This study underscores the growing burden of MDRB and highlights the importance of enhanced antimicrobial stewardship in respiratory ICUs.

## 1. Introduction

Patients admitted to intensive care units (ICUs) are at high risk of infections due to their critical conditions, which often require intensive treatments and invasive procedures. In these patients, infections caused by multidrug-resistant (MDR) bacteria frequently arise as a consequence of extensive antibiotic use. These bacteria significantly limit treatment options, extend hospital stays, and increase mortality rates [1].

Antibiotic resistance has emerged as a growing threat to global healthcare systems. The increasing prevalence of MDR bacteria not only compromises patient outcomes but also imposes substantial financial burdens on healthcare services. Identifying effective strategies to combat these infections in ICUs is, therefore, of paramount importance [2]. However, the prolonged and widespread use of broad-spectrum antibiotics to treat MDR infections accelerates the emergence of resistant bacteria [3].

Culture positivity is a vital biomarker for assessing the risk of infection among ICU patients. Prolonged exposure to treatments such as invasive procedures and mechanical ventilation facilitates the colonization of Gram-negative bacteria and MDR organisms. Patients with culture positivity are at higher risk of developing severe infections and often require readmission. Additionally, culture positivity plays a crucial role in the early detection of pathogens, which aids in preventing infections and guiding treatment strategies [4].

This study aims to investigate the relationships between culture positivity, antibiotic resistance, associated risk factors, readmission rates, and clinical outcomes in patients admitted to respiratory ICUs. By evaluating the impact of culture positivity and antibiotic resistance on patient prognosis, this study seeks to identify effective strategies for managing infections and improving clinical outcomes.

One of the innovative aspects of this study lies in its comprehensive evaluation of culture positivity and multidrug-resistant bacteria (MDRB) in respiratory ICU patients, not only in terms of infection burden but also in relation to readmission rates and survival outcomes. Furthermore, the interpretation of culture positivity as indicative of both active infection and colonization—supported by laboratory parameters—offers a novel perspective for infection control strategies. In this regard, the study moves beyond microbiological outcomes alone and adopts a holistic approach by integrating multiple clinical variables that influence patient management and ICU processes.

## 2. Materials and Methods

This study was conducted in compliance with the ethical standards of the institutional and national research committee and adhered to the 1964 Helsinki Declaration and its later amendments. Informed consent was obtained from all participants included in the study, along with consent to publish the study findings.

The study was designed as a retrospective, cross-sectional investigation and conducted in a tertiary care hospital specializing in pulmonary medicine. Ethics approval was obtained from the Clinical Research Ethics Committee of the University of Health Sciences, Ankara Sanatorium Training and Research Hospital (approval date: 13 November 2024, approval number: 2024-BÇEK/164). The study population consisted of patients admitted to the pulmonary intensive care unit (ICU) between 1 January 2019, and 1 January 2020. All included patients had at least one primary respiratory comorbidity that justified their ICU admission. Patients were not enrolled in a specific follow-up program; however, their survival status after discharge was assessed via the national healthcare database.

Demographic data, including gender, age, comorbidity history, intubation history, previous culture results, and past ICU admissions, were collected from the hospital database. For cases where complete information was unavailable, the national patient database was consulted, especially when patients had been admitted to other healthcare centers prior to the current ICU admission. Additional information, such as pneumonia at admission, pulmonology consultations, source of ICU admission (e.g., outpatient clinic, emergency ward, inpatient ward, or another ICU), non-invasive mechanical ventilation (NIMV) use, and antibiotic regimens, was extracted from ICU records. Antibiotherapy regimens were categorized into three groups based on respiratory severity: 1—combinations involving cephalosporins, penicillins, macrolides, or fluoroquinolones; 2—antipseudomonal combinations; and 3—carbapenems and other broad-spectrum regimens. Pathogen-specific treatments, such as antifungals, antivirals, and methicillin-resistant *Staphylococcus aureus* (MRSA) treatments, were recorded separately.

Culture results were considered positive for a specific bacterium regardless of the sample origin, as long as the result was not deemed contaminated. Samples were classified into respiratory (e.g., endotracheal aspiration, orofacial sampling, sputum), urinary, or hematological (e.g., catheter and blood cultures) categories. For patients with multiple positive cultures, the most resistant sample was reported. If the same resistance pattern was reported across multiple samples, the earliest reported result was used. Cultures were classified as multidrug-resistant if at least one result indicated resistance to commonly used regimens. Rare culture samples (less than 1% of cases) were excluded from the statistical analysis.

Survival duration was calculated from the date of the initial ICU admission. For patients with repeated ICU admissions, survival was assessed based on the initial admission. ICU mortality (exitus) was defined as death during the ICU stay, regardless of whether it occurred during the initial or subsequent admission. ICU length of stay was defined as the time from admission to discharge or, for deceased patients, to the time of death during the initial admission. Durations of repeated admissions were excluded from statistical evaluation.

Laboratory markers were collected at different time points. Results obtained at admission were considered baseline values. Follow-up values were recorded as part of routine weekly tests, with the first follow-up conducted 3–4 days after admission and the second follow-up 6–8 days after admission. For cases with additional tests due to clinical necessity, values closest to these time points were used. Laboratory results at or near discharge were accepted as the third and final follow-up values.

### Statistical Analysis

All collected data were initially organized and reviewed in Microsoft Excel to ensure accuracy and completeness. Statistical analyses were performed using IBM SPSS Statistics for Windows (Version 25.0, IBM Corp., Armonk, NY, USA). Descriptive analyses were conducted, with results presented as mean and standard deviation for normally distributed data or as median and percentiles for non-normally distributed data. Parametric distribution was evaluated using Q-Q plot analysis.

For comparisons, Chi-square tests and independent samples t-tests were used for nominal and scale variables with parametric distributions, respectively. For non-parametric data, the Mann–Whitney U test was applied. Chi-square analyses excluded parameters with subgroup counts below expected thresholds to ensure reliable comparisons. Statistical significance was set at *p* ≤ 0.05.

To establish a valid model for linear bivariate regression analysis with a Type 1 error rate of 0.05 and 90% power, a minimum of 190 patients was required, assuming an equal allocation ratio between groups. The study design, however, anticipated a sample size at least twice this number to allow for more robust subgroup evaluations.

## 3. Results

A total of 695 patient admissions were evaluated for the study. Of these, 24 records were excluded: 10 patients due to confirmed outlier data for laboratory markers, 4 patients due to hospital system limitations resulting in incomplete demographic information, and 10 patients due to indeterminate culture results that prevented an exact diagnosis. These 24 excluded patients were not included in any statistical analyses or tables. Ultimately, 519 unique patient records were included in the analysis, with an additional 152 records evaluated for readmissions. Among these 152 records, 65 unique patients accounted for 87 repeat admissions. Patients were first grouped based on their readmission status into “readmitted” (at least one readmission) and “no readmission”. Subsequently, patients were reclassified based on culture results and multidrug resistance (MDR) (Figure 1).

Table 1 presents the demographic and clinical characteristics of the study population. The mean age of patients with readmissions and those without was 69.6 (±11.48) and 69.97 (±11.9) years, respectively. Male patients were more prevalent in the readmission group (*n* = 48, 73.8%) compared to the no-readmission group (*n* = 294, 56.6%), and this gender difference was statistically significant (*p* = 0.008). Cardiovascular disease was the most common comorbidity, affecting 60% (*n* = 39) of readmitted patients and 66.4% (*n* = 344) of non-readmitted patients. Neurological comorbidities and malignancy were rare, with 7.7% (*n* = 40) of non-readmitted patients and 1.6% (*n* = 1) of readmitted patients affected. Pneumonia at admission was present in approximately one-third of patients in both groups. Emphysema was observed in less than 1% of patients, while bronchiectasis was more frequent in the readmission group (6.2%) compared to the no-readmission group (3.3%).

More than half of ICU admissions in both groups originated from other ICUs, but patients with readmissions were more likely to be admitted from emergency wards compared to those without readmissions (46.2% vs. 33%, *p* = 0.038). Intubation history within the past year was similar between the groups (12.3% vs. 11%), as was the need for non-invasive mechanical ventilation (NIMV) (79% in the readmission group vs. 67.9% in the no-readmission group, *p* = 0.072). The most common first-line treatment for both groups was a combination of cephalosporins with macrolides or fluoroquinolones (44.6% vs. 43.9%), followed by antipseudomonal regimens (32.3% vs. 24.5%), with no statistically significant difference in treatment preferences (*p* = 0.389) (Table 1). Additional treatment regimens, such as antifungal and antiviral therapies, were required in less than 5% of patients in both groups. MRSA-targeted treatment was used in 4.6% of the readmission group and 7.3% of the no-readmission group.

As shown in Table 2, culture positivity and MDR status were analyzed in relation to ICU outcomes. Positive culture samples were most frequently derived from respiratory sources (57.7% in the readmission group vs. 59.8% in the no-readmission group), followed by urinary (38.5% vs. 33.7%) and hematological samples, with no significant difference between groups (*p* = 0.426). Patients with multiple positive cultures (34.6% vs. 38.5%) and those with MDR cultures (65.4% vs. 61.5%) were similar between groups. *Klebsiella *spp. was the most common isolate in both groups (23.1% vs. 21.1%), followed by *Candida* spp. (23.1% vs. 11.1%) and *Pseudomonas* spp. (15.4% vs. 11.1%). No significant differences in overall culture results were observed between the groups (*p* = 0.240). However, culture positivity within six months prior to ICU admission was more frequent in the readmission group (12.5%, *n* = 8) compared to the no-readmission group (3.7%, *n* = 11, *p* = 0.033). Resistance patterns among these cultures were similar between the groups.

The median ICU length of stay was 8 days [5,6,7,8,9,10,11] for the readmission group and 7 days [5,6,7,8,9,10,11] for the no-readmission group. Survival duration was significantly longer in the readmission group (median 238 days) compared to the no-readmission group (median 59 days, *p* = 0.047, log-rank Mantel–Cox survival analysis). ICU mortality (exitus) did not differ significantly between the groups (Table 2).

### 3.1. MDR vs. Non-MDR Comparison

Table 3 provides a comparison of MDR patterns with demographic and clinical parameters of the study population. No significant differences were observed between patients with MDR and non-MDR cultures regarding age (72.54 vs. 70.21 years), gender distribution, or comorbidities. Admission origin was also similar, with the majority of patients in both groups admitted from other ICUs (75.9% vs. 78.9%). Intubation history and NIMV requirements were comparable (24.1% vs. 19.7%, *p* = 0.449; 77% vs. 67.6%, *p* = 0.15). Among culture subgroups, *Acinetobacter *spp. (18.4% vs. 3.8%) and *Klebsiella *spp. (28.6% vs. 8.9%) were significantly more prevalent in the MDR group, while *Candida *spp. was predominantly found in the non-MDR group (32.9% vs. 0.7%, *p* = 0.001). The median ICU length of stay was longer for the MDR group (10 days, 6–17) compared to the non-MDR group (8 days, 5–12, *p* = 0.013). Survival duration did not differ significantly between MDR and non-MDR groups (30 days, 12–134 vs. 41 days, 16–147, *p* = 0.739, Figure 2 and Figure 3) (Table 3).

### 3.2. Laboratory Values

Table 4 summarizes the laboratory and readmission-related parameters. Admission laboratory parameters, including white blood cell (WBC) count, neutrophil count (NEU), C-reactive protein (CRP), and procalcitonin (PRC), did not differ significantly between the readmission and no-readmission groups. However, readmitted patients had significantly lower CRP levels at admission (21 mg/dL, 11.5–60.6) compared to non-readmitted patients (35 mg/dL, 14.04–93.28, *p* = 0.023). Similarly, at the final follow-up, CRP levels remained lower in the readmission group (12 mg/dL) compared to the no-readmission group (28.06 mg/dL, *p* = 0.004). WBC counts were also significantly lower at the final follow-up in the readmission group (8.60 × 10^9^/L) compared to the no-readmission group (9.61 × 10^9^/L, *p* = 0.041). No other significant differences were noted in the first and second follow-up evaluations.

For MDR and non-MDR comparisons, no significant differences were observed in WBC, NEU, CRP, or PRC levels during admission or follow-up evaluations (Table 4).

## 4. Discussion

### 4.1. Gender and Admission Source as Readmission Predictors

In our study, we aimed to reveal the differences in clinical, laboratory, and infectious processes between patients with ICU readmissions and those admitted for the first time. Some of our findings were particularly striking and may contribute to the existing literature. For instance, we observed that male gender was significantly associated with ICU readmissions (*p* = 0.008). A gender-based ICU study conducted by Blecha et al. in 2020 reported that male patients were more likely to undergo tracheostomy, receive dialysis, and spend significantly more time on mechanical ventilation compared to female patients [5]. Based on these findings, it can be inferred that male patients admitted to the ICU are subjected to more invasive procedures and have a higher likelihood of ICU readmissions compared to females.

Similarly, Ponzoni et al. (2017) investigated the risk factors associated with ICU readmissions and found that patients who were admitted to the ICU as part of a planned postoperative transfer had lower rates of readmission compared to those admitted from general wards or emergency departments [6]. Consistent with this literature, our study also found that patients admitted from the emergency department had significantly higher rates of ICU readmissions during their hospital stay (*p* = 0.038).

In the groups with and without ICU readmissions, no significant differences were found in the need for non-invasive or invasive mechanical ventilation. However, contrary to the literature, the survival durations of patients with ICU readmissions were significantly higher. In Ponzoni’s study, ICU readmissions were associated with significantly higher ICU mortality, 90-day mortality, and overall hospital mortality [6]. Similarly, a study by Hanberger et al. demonstrated that infections caused by difficult-to-treat pathogens, such as *Pseudomonas aeruginosa*, *Acinetobacter *spp., and *MRSA*, were generally associated with higher ICU and hospital mortality rates compared to other infections. Furthermore, high antibiotic resistance (HighABR) was shown to be significantly more prevalent in ICUs [4,7]. Other ICU studies indicate that the absence of appropriate empirical treatments for these pathogens is linked to an increase in attributable mortality [7,8,9,10].

### 4.2. Microbiological Profile and Pathogen Distribution

Our patient population predominantly consisted of a relatively homogeneous group with respiratory pathologies. Chronic and/or acute exacerbations of diseases such as COPD, IPF, asthma, and lung cancer were prominent, all of which could lead to ICU readmissions. We believe that the effective treatment and stabilization of exacerbations under ICU conditions extended survival. However, no significant differences in 2-year mortality rates were found between the groups.

In both the readmitted and non-readmitted groups, the most frequently isolated microorganisms were from respiratory tract samples. In both groups, *Klebsiella *spp. was the leading pathogen, exceeding 20% of isolates, followed by *Candida *spp. and *Pseudomonas *spp. When examining the rates of multidrug-resistant bacteria (MDRB), we found that MDRB prevalence exceeded 60% in both groups. Analyzing MDRBs revealed that *Klebsiella *spp. was dominant in both groups, followed by *Acinetobacter *spp. A study conducted by Ibrahim et al. in Libya showed that out of 197 swabs collected from ICU patients, staff, and equipment, the most commonly isolated microorganisms were *Acinetobacter *spp. (44%) and *Klebsiella *spp. (40%) [11]. Moreover, a large multicenter study involving 57 ICUs across 24 countries identified *Klebsiella *spp. as the leading MDRB in patients with bacteremia, while *Acinetobacter *spp. was prominent in the extensively drug-resistant (XDR) profile [12].

Indeed, the prevalence of *Acinetobacter *spp. strains has increased in recent years, with these pathogens becoming resistant to broad-spectrum antibiotics commonly used in routine care. Additionally, they have the capacity to develop new resistance mechanisms during treatment [13,14]. Similarly, a study by Hasanzade et al. involving 287 respiratory ICU patients reported that 55.1% of cultured isolates were Gram-negative bacteria, while 25.1% were Gram-positive bacteria. Among these isolates, *Acinetobacter *spp. showed resistance to over 95% of antibiotics except colistin [15].

### 4.3. Clinical Impact of MDRB on ICU Outcomes

In our study, patients with ICU readmissions had higher positive culture rates within the past six months compared to non-readmitted patients. As previously noted, considering that more than 50% of these cultures were isolated from respiratory tract samples, this finding can be attributed to acute infectious exacerbations of respiratory system diseases or uncontrolled chronic infections.

When comparing patients with MDRB-positive and MDRB-negative cultures, as expected, ICU stays were significantly longer in the MDRB-positive group compared to the MDRB-negative group (*p* = 0.013), consistent with trends observed in ICUs worldwide. However, no significant difference was found in survival durations between the two groups. A 2020 study from Taiwan analyzed 60,317 ICU patients and compared 279 patients with MDRB-positive cultures to an equal number of patients without MDRB. The study found that ICU length of stay increased by 13%, overall hospital costs were 26% higher for the MDRB group, and healthcare service (8%), medication (26.9%), and bed (10.3%) costs were significantly elevated. Despite these differences, there was no significant difference in mortality rates between MDRB and non-MDRB groups [16].

Both the literature and our study demonstrate that patients infected with MDRB incur prolonged ICU stays and increased treatment costs, significantly impacting healthcare policies and decision-making. In Turkey, high rates of antibiotic resistance have compelled healthcare administrators to take measures to curb the prescription of unnecessary and inappropriate antibiotics, even at the primary care level.

### 4.4. Laboratory Biomarkers and the Role of Colonization

One of the intriguing findings in our study pertained to laboratory parameters. We observed that ICU admission and discharge CRP levels in the ICU readmission group were significantly lower than in the non-readmission group (*p* = 0.004). Similarly, white blood cell counts at ICU admission were also lower in the readmission group (*p* = 0.041). In contrast, most studies in the literature show high CRP and white blood cell levels at ICU discharge with significantly increased rates of ICU readmission [17,18]. Procalcitonin is acknowledged as a more specific biomarker than C-reactive protein for detecting bacterial inflammation [19]. However, procalcitonin did not differ in our study groups.

Considering that the readmitted group in our study predominantly consisted of patients with chronic diseases experiencing acute exacerbations and given the significantly higher culture positivity rates within the past six months compared to the non-readmission group, it is plausible that a substantial proportion of MDRB-positive cultures in this group represent colonization rather than active infection. This may explain the statistically lower CRP levels and longer survival durations in the readmission group.

According to the literature, the prevalence of MDRB colonization in ICU patients is reported to be 25.3%, with an incidence of 45.14% [18].

Recent studies reinforce the clinical importance of colonization in ICU settings, particularly in the context of MDR bacteria. Both Heath et al. and a recent observational study by Gracheva et al. emphasize that the primary focus in colonized patients should be on prevention rather than treatment. The latter also highlights that monomicrobial culture positivity is more likely to progress to infection compared to polymicrobial findings, suggesting that not all colonization carries the same risk [20,21]. Therefore, colonization is by no means clinically irrelevant. On the contrary, it is of high importance—but preventing it before it develops and accurately distinguishing it from active infection when present are crucial steps in avoiding unnecessary and excessive antibiotic use.

### 4.5. Future Research Directions

Our study offers a multidimensional perspective on the clinical impact of culture positivity and multidrug-resistant bacteria (MDRB) in patients admitted to respiratory intensive care units. The observation that MDRB prevalence was high in both readmitted and non-readmitted patients, despite similar mortality rates, suggests that colonization may play a more prominent role than active infection in this population. This finding challenges conventional infection control strategies and underscores the need to revisit diagnostic criteria that distinguish clinically significant infections from mere colonization. Furthermore, while MDRB positivity was associated with prolonged ICU stays, it did not correspond with increased mortality, indicating that these organisms may impose a greater burden on healthcare resources rather than directly affecting patient survival. The lower CRP and WBC levels observed in readmitted patients, along with their longer survival times, support the hypothesis that culture positivity in this group may reflect chronic colonization and subsequent exacerbations rather than acute, life-threatening infections. However, although colonization may not always require immediate antimicrobial treatment, it still warrants careful monitoring and individualized clinical evaluation, especially in patients with frequent exacerbations or immunocompromised status. Future research should focus on developing predictive models incorporating laboratory trends, colonization history, and readmission patterns to improve antimicrobial stewardship and optimize ICU resource utilization.

## 5. Limitations of the Study

The primary limitation of our study is its retrospective design. Furthermore, since the study was conducted in a relatively homogeneous patient group, with a significant proportion of culture positivity results derived from the respiratory system, the findings may not be generalizable to all ICU patients.

## 6. Conclusions

Infectious outbreaks that result in substantial patient mortality within a short time frame have always received well-deserved attention in the medical community. However, bacterial infections that are challenging to treat—encountered daily by physicians during inpatient ward rounds or outpatient visits—not only account for significant annual patient mortality but also impose a heavy financial burden on healthcare systems.

In ICUs, infectious disease specialists and intensivists are increasingly discussing cases where no effective antibiotics remain for the microorganism affecting a patient. Patients with prolonged ICU stays, prior exposure to multiple antibiotic treatments, and chronic diseases colonized by multidrug-resistant microorganisms pose a significant risk to newly admitted ICU patients. These scenarios are fostering a shift in perspective among intensivists, where ICU admissions may be perceived as causing more harm than benefit in certain cases.

This study underscores the growing visibility of what has long been the “hidden part of the iceberg”. Addressing these challenges urgently requires targeted healthcare policies and interventions to mitigate the escalating impact of multidrug-resistant bacteria in ICUs.

## Figures and Tables

**Figure 1 diagnostics-15-01737-f001:**
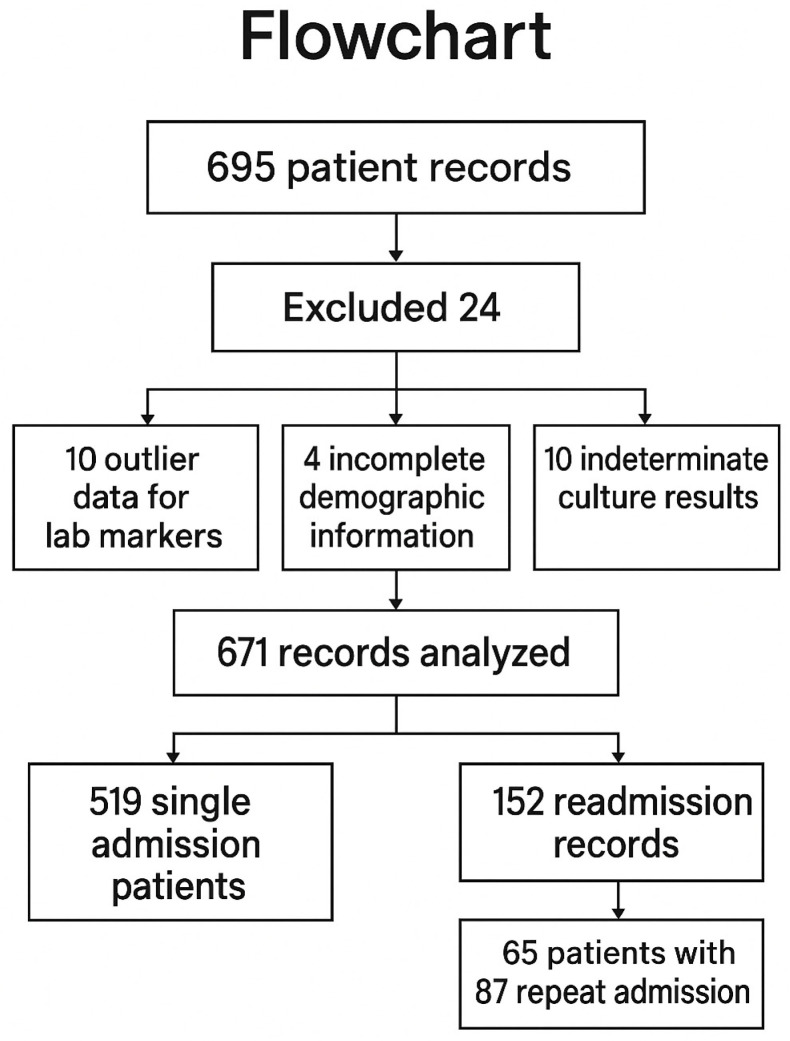
Flowchart of the study.

**Figure 2 diagnostics-15-01737-f002:**
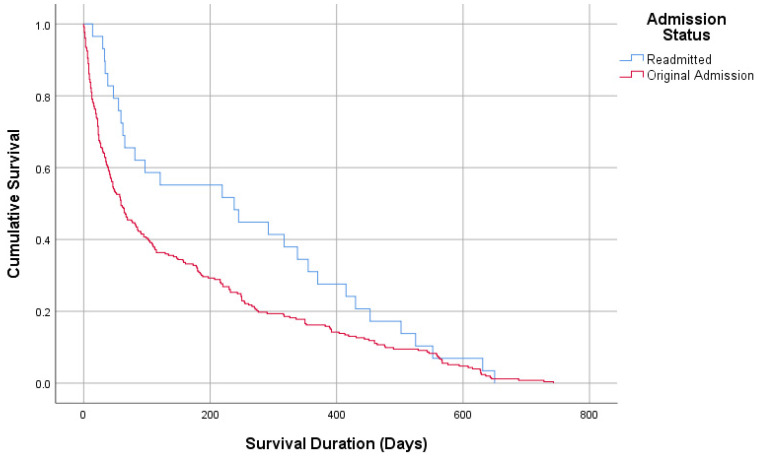
Survival analysis according to admission.

**Figure 3 diagnostics-15-01737-f003:**
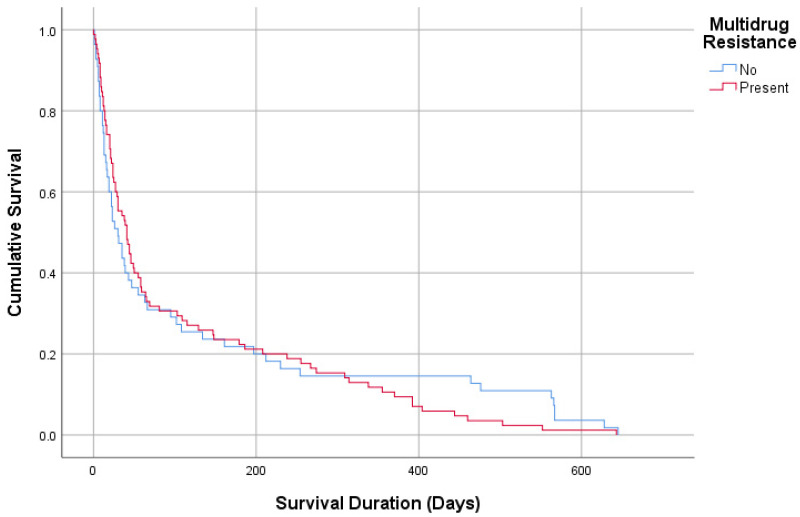
Survival analysis according to multidrug resistance.

**Table 1 diagnostics-15-01737-t001:** Demographic parameters, comorbidities, and treatment regimen.

Parameters (*n*, %)		Readmitted (*n* = 65)	No Readmission (*n* = 519)	*p* Value
Age (years, ± SD)		69.6 ± 11.48	69.97 ± 11.9	0.812
Gender	Male	48 (73.8)	294 (56.6)	0.008 *
	Female	17 (26.2)	225 (43.4)	
Comorbidities				
Cardiovascular	None	26 (40)	174 (33.6)	0.305
	Present	39 (60)	344 (66.4)	
Metabolic	None	49 (75.4)	334 (64.5)	0.081
	Present	16 (24.6)	184 (35.5)	
Neurological	None	63 (98.4)	478 (92.3)	N/A
	Present	1 (1.6)	40 (7.7)	
Malignancy	None	63 (98.4)	478 (92.3)	N/A
	Present	1 (1.6)	40 (7.7)	
Respiratory Comorbidities				
Pneumonia at Admission	No	44 (67.7)	338 (65.1)	0.682
	Yes	21 (32.3)	181 (34.9)	
Bronchiectasis	None	61 (93.8)	502 (96.7)	N/A
	Present	4 (6.2)	17 (3.3)	
Emphysema	None	65 (100)	516 (99.4)	N/A
	Present	0 (0.0)	3 (0.6)	
ICU Origin of Admission ^1^	Outpatient	0 (0.0)	3 (0.6)	0.038 *
	Emergency Ward	30 (46.2)	171 (33.0)	
	Inpatient Ward	2 (3.1)	17 (3.3)	
	Other ICU	33 (50.8)	327 (63.1)	
Intubation History within a year	No	57 (87.7)	462 (89)	N/A
	Yes	8 (12.3)	57 (11)	
NIV Requirement in ICU Setting	No	13 (21)	163 (32.1)	0.072
	Required	49 (79)	344 (67.9)	
Antibiotherapy Regimens	None	6 (9.2)	54 (10.4)	0.389
	Cephalosporin/Macrolide or Fluoroquinolone Combination	29 (44.6)	228 (43.9)	
	Antipseudomonal Combination	21 (32.3)	127 (24.5)	
	Carbapenem and Other Broad Spectrum	9 (13.8)	110 (21.2)	

SD: Standard Deviation, ICU: Intensive Care Unit, NIMV: Non-invasive Mechanical Ventilation. ^1^ The Chi-Square test was performed between the “Emergency Ward” and “Other ICU” groups. Values are presented as mean ± standard deviation (SD) for continuous variables and number (percentage) for categorical variables. * significant *p* values, N/A: Not applicable

**Table 2 diagnostics-15-01737-t002:** Culture results and survival evaluation.

Parameters (*n*, %)		Readmitted (*n* = 65)	No Readmission (*n* = 519)	*p* Value
Additional Treatment Requirements				
Antifungal	No	63 (96.9)	507 (97.7)	N/A
	Required	2 (3.1)	12 (2.3)	
Antiviral	No	63 (96.9)	497 (95.8)	N/A
	Required	2 (3.1)	22 (4.2)	
*MRSA*	No	62 (95.4)	481 (92.7)	N/A
	Required	3 (4.6)	38 (7.3)	
Origin of Culture	Respiratory	15 (57.7)	101 (59.8)	0.426
	Urinary	10 (38.5)	57 (33.7)	
	Blood	1 (3.8)	11 (6.5)	
Multidrug Resistance in Culture	No	10 (38.5)	64 (37.9)	0.954
	Present	16 (61.5)	105 (62.1)	
More than One Positive Culture Result	No	17 (65.4)	104 (61.5)	N/A
	Yes	9 (34.6)	65 (38.5)	
Culture Result		Readmitted (*n* = 26)	No Readmission (*n* = 171)	
Other		5 (19.2)	21 (12.3)	0.240
*Acinetobacter* spp.		2 (7.7)	27 (15.8)	
*Candida* spp.		6 (23.1)	19 (11.1)	
*Enterococcus* spp.		0 (0.0)	14 (8.2)	
*Escherichia coli*		3 (11.5)	20 (11.7)	
*Klebsiella* spp.		6 (23.1)	36 (21.1)	
*Staphylococcus* spp.		0 (0.0)	15 (8.8)	
*Pseudomonas* spp.		4 (15.4)	19 (11.1)	
Culture Positivity within 6 Months	No	56 (87.5)	489 (94.4)	0.033 *
	Present	8 (12.5)	29 (5.6)	
Resistant Culture within 6 Months	No	3 (37.5)	11 (37.9)	N/A
	Present	5 (62.5)	18 (62.1)	
ICU Admission Duration (Days, Median, 25–75th percentile)		8 (5–11)	7 (5–11)	0.977
Survival Duration (Days, Median, 25–75th percentile)		238 (59–415)	59 (20–243)	0.047 ^1^/0.002 ^2^
Exitus	No	36 (55.4)	266 (51.3)	0.530
	Yes	29 (44.6)	253 (48.7)	
Exitus within ICU Admission	No	64 (98.5)	492 (94.8)	N/A
	Yes	1 (1.5)	27 (5.2)	

MRSA: Methicillin-resistant staphylococcus aureus, ICU: Intensive Care Unit; ^1^ Log Rank Mantel–Cox and ^2^ Mann–Whitney U tests were utilized for survival evaluation. * significant *p* values, N/A: Not applicable

**Table 3 diagnostics-15-01737-t003:** Comparison of multidrug resistance and demographic parameters.

Parameters (*n*, %)		Non-MDR (*n* = 79)	MDR (*n* = 147)	*p* Value
Age (years, SD)		72.54 (11.3)	70.21 (11.13)	0.138
Gender	Male	42 (53.2)	86 (58.5)	0.44
	Female	37 (46.8)	61 (41.5)	
Comorbidities				
Cardiovascular	None	31 (29.2)	47 (32)	0.273
	Present	48 (60.8)	100 (68)	
Metabolic	None	43 (54.4)	99 (67.3)	0.055
	Present	36 (45.6)	48 (32.7)	
Pneumonia at Admission	No	49 (62)	76 (51.7)	0.137
	Yes	30 (38)	71 (48.3)	
ICU Origin of Admission ^1^	Outpatient	0 (0.0)	1 (0.7)	0.069
	Emergency Ward	16 (20.3)	24 (16.3)	
	Inpatient Ward	3 (3.8)	6 (4.1)	
	Other ICU	60 (75.9)	116 (78.9)	
Intubation History within a Year	No	60 (75.9)	118 (80.3)	0.449
	Yes	19 (24.1)	29 (19.7)	
NIMV Requirement in ICU Setting	No	17 (23)	45 (32.4)	0.15
	Required	57 (77)	94 (67.6)	
Culture Result				
Other		13 (16.5) a	15 (10.2) a	0.001 *
*Acinetobacter* spp.		3 (3.8)	27 (18.4) b	
*Candida* spp.		26 (32.9) a	1 (0.7) b	
*Enterococcus* spp.		5 (6.3) a	14 (9.5) a	
*Escherichia coli*		9 (11.4) a	20 (13.6) a	
*Klebsiella* spp.		7 (8.9) a	42 (28.6) b	
*Staphylococcus* spp.		9 (11.4) a	7 (4.8) a	
*Pseudomonas* spp.		7 (8.9) a	21 (14.3) a	
ICU Admission Duration (Days, Median, 25–75th percentile)		8 (5–12)	10 (6–17)	0.013 *
Survival Duration (Days, Median, 25–75th percentile)		30 (12–134)	41 (16–147)	0.739 ^2^/0.375 ^3^
Exitus	No	24 (30.4)	62 (42.2)	0.082
	Yes	55 (69.6)	85 (57.8)	
Exitus within ICU Admission	No	70 (88.6)	139 (94.6)	0.106
	Yes	9 (11.4)	8 (5.4)	

MDR: Multidrug Resistance, SD: Standard Deviation, MRSA: Methicillin-resistant staphylococcus aureus, ICU: Intensive Care Unit; ^1^ The Chi-Square test was performed between the “Emergency Ward” and “Other ICU” groups. Pairwise comparisons were made between groups, with different groups being stated as “a” and “b”. ^2^ Log Rank Mantel–Cox and ^3^ Mann–Whitney U Test were utilized for survival evaluation. * significant *p* values.

**Table 4 diagnostics-15-01737-t004:** Laboratory results during admission and follow-up.

Parameters		Readmitted (*n* = 65)	No readmission (*n* = 519)	df	Mean Difference	*p* Value
White Blood Cell Count (×10^9^/L)	Admission	10.5 (4.1)	11.03 (4.45)	549	−0.53	0.38
	First Follow-up	8.86 (3.52)	9.43 (3.85)	543	−0.57	0.269
	Second Follow-up	8.60 (2.91)	9.57 (3.8)	422	−0.97	0.08
	Third Follow-up	8.00 (2.63)	9.61 (4.03)	231	−1.61	0.041
Neutrophile Count (×10^9^/L)	Admission	8.48 (3.37)	8.67 (3.89)	547	−0.18	0.722
	First Follow-up	6.92 (3.02)	7.11 (3.1)	539	−0.19	0.642
	Second Follow-up	6.20 (2.67)	7.02 (3.1)	412	−0.83	0.87
	Third Follow-up	6.47 (3.68)	6.92 (3.28)	228	−0.44	0.583
				** *n* **	**Z score**	***p* value**
C-Reactive Protein (mg/dL)	Admission	21 (11.5–60.6)	35 (14.04–93.28)	543	2.281	0.023 *
	First Follow-up	21.6 (10–53)	25.59 (11–70.18)	530	1.015	0.31
	Second Follow-up	19.3 (6.81–41.87)	22.5 (9.4–58.25)	389	1.326	0.185
	Third Follow-up	12 (6.4–12)	28.06 (12.6–70.35)	207	2.894	0.004 *
Procalcitonin (ng/mL)	Admission	0.11 (0.05–0.25)	0.1 (0.05–0.24)	306	−0.162	0.872
	Follow-up	0.06 (0.05–0.29)	0.14 (0.07–0.32)	123	0.835	0.403
**Parameters**		**Non-MDR (*n* = 79)**	**MDR (*n* = 147)**	**df**	**Mean Difference**	***p* Value**
White Blood Cell Count (×10^9^/L)	Admission	11.59 (5.16)	10.92 (4.47)	211	0.67	0.321
	First Follow-up	9.83 (3.82)	9.27 (4.08)	208	0.55	0.334
	Second Follow-up	10.12 (4.27)	9.08 (3.38)	166	1.03	0.09
	Third Follow-up	10.36 (4.38)	8.87 (3.49)	80	1.49	0.096
Neutrophile Count (×10^9^/L)	Admission	8.80 (4.30)	9.01 (3.76)	201	−0.21	0.718
	First Follow-up	6.76 (3.04)	7.50 (3.44)	199	−0.73	0.134
	Second Follow-up	7.10 (2.69)	7.38 (3.27)	185	−0.27	0.546
	Third Follow-up	7.06 (3.12)	7.06 (3.49)	127	−0.01	0.996
				** *n* **	**Z score**	***p* value**
C-Reactive Protein (mg/dL)	Admission	56.6 (21–133)	57.5 (24–101.88)	201	−0.59	0.555
	First Follow-up	52 (17.84–84.53)	44 (17.65–92.04)	198	0.167	0.868
	Second Follow-up	34 (13.25–81.78)	32.7 (12.92–70.5)	181	−0.16	0.873
	Third Follow-up	37.5 (11.09–74.62)	37.03 (15.72–74.7)	116	−0.069	0.945
Procalcitonin (ng/mL)	Admission	0.16 (0.07–0.49)	0.11 (0.07–0.44)	155	−0.283	0.778
	Follow-up	0.20 (0.06–0.57)	0.22 (0.11–0.39)	74	0.206	0.837

df: Degree of Freedom, Z Score: Standardized Test Statistic, Independent Samples T-test was utilized for white blood cell and neutrophil comparison between groups, while Mann–Whitney U Test was used for comparison regarding C-reactive protein and procalcitonin due to nonparametric distribution. * significant ***p*** values.

## Data Availability

The datasets generated and/or analyzed during the current study are available from the corresponding author upon reasonable request.

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
