# Peer review of "Culture Positivity and Antibiotic Resistance in Respiratory Intensive Care Patients: Evaluation of Readmission and Clinical Outcomes"

_diagnostics, 2025, doi:10.3390/diagnostics15141737_

Round 1
Reviewer 1 Report
Comments and Suggestions for Authors
Discuss the implications/ future directions of your study findings,
Eg ICU readmissions with gender
Emphasize the novelty of your study
Format the discussion giving more priority to findings related to the objective of the study
Author Response
Response to Reviewer 1 Comments
Comment 1:
Discuss the implications/ future directions of your study findings, e.g., ICU readmissions with gender.
Response:
We appreciate the reviewer’s insightful suggestion. We have expanded the Discussion section by adding a dedicated subsection titled '4.5. Future Research Directions'. In this section, we interpret our findings in the context of future clinical and research implications, especially regarding the potential role of colonization in culture-positive patients and the need for new predictive models incorporating colonization history and lab markers like CRP and WBC. This addresses the broader impact of our findings beyond gender and aligns with the study’s multidimensional scope.
Comment 2:
Emphasize the novelty of your study.
Response:
Thank you for this valuable recommendation. To emphasize the novelty of our study, we added a paragraph at the end of the Introduction. This paragraph highlights our study's unique aspects, such as evaluating culture positivity and MDRB not only in relation to infection burden but also with respect to ICU readmissions and survival durations. Additionally, it emphasizes our analytical approach to differentiating between colonization and active infection using laboratory parameters.
Comment 3:
Format the discussion giving more priority to findings related to the objective of the study.
Response:
In accordance with the reviewer’s suggestion, we restructured the Discussion section by dividing it into clearly labeled subsections such as '4.1 Gender and Admission Source as Readmission Predictors', '4.2 Microbiological Profile and Pathogen Distribution', '4.3 Clinical Impact of MDRB on ICU Outcomes, 4.5 Future Research Directions and so on. This format enhances clarity and ensures that our findings are directly aligned with the primary objectives of the study.
Reviewer 2 Report
Comments and Suggestions for Authors
justify this at abstract Lower significance means that, but even positive lower means possible inflamation
Eliminate this comment Line 43 further exacerbating the issue
How do you think consider that ? Means that they do not need atennd aor new treatment ?
this group represent colonisation rather than active infection do they not need assistance ?
Author Response
Response to Reviewer 2 Comments
Comment 1:
Justify this at abstract: 'Lower significance means that, but even positive lower means possible inflammation'
Response:
Thank you for pointing this out. We clarified this statement in the abstract. It now explains that although CRP levels in readmitted patients were above the normal threshold, they were significantly lower than in non-readmitted patients. This pattern may reflect subclinical inflammation associated with colonization rather than active infection, particularly in patients with chronic respiratory conditions.
Comment 2:
Eliminate this comment: Line 43 'further exacerbating the issue'
Response:
We agree with the reviewer’s observation. The phrase 'further exacerbating the issue' has been removed from the manuscript for clarity and precision.
Comment 3:
How do you think consider that? Means that they do not need attend or new treatment?
Response:
We appreciate the reviewer’s concern. We clarified in the discussion that although colonized patients may not always require immediate antimicrobial treatment, they still warrant careful monitoring and individualized evaluation—especially those with frequent exacerbations or immunocompromised conditions.
Comment 4:
This group represent colonisation rather than active infection – do they not need assistance?
Response:
Thank you for this important point. We further elaborated in the discussion by referencing recent studies (Heath et al., 2024; Gracheva et al., 2023), both of which emphasize that colonization, particularly with MDR organisms, should not be underestimated. While immediate treatment may not always be warranted, prevention, risk stratification, and careful clinical distinction from active infection are essential to avoid inappropriate antibiotic use. We have added a paragraph highlighting that monomicrobial cultures may pose a higher risk of progression to infection and that colonization requires clinical vigilance.
Reviewer 3 Report
Comments and Suggestions for Authors
The peer-reviewed manuscript is a retrospective, epidemiological study conducted in a tertiary care hospital specializing in pulmonary medicine. The authors analyzed complex data including demographic data (gender, age, comorbidity history, intubation history, previous culture results, and past ICU admissions) in the context of patients’ ICU stay.
The following comments were made regarding the text of the manuscript:
1. L. 123-131: When describing the results, there is no clear understanding of the formation of the two groups of patients; therefore, the Materials and Methods section should include a schematic design for the patients who were included in the study and a rationale for those patients who were excluded from the study.
2. L. 129: Clarify whether these patients were included at some point in the study, since these data are not presented in the tables.
3. Provide reference to all tables at the beginning of their description.
4. Table 1: Provide a designation for the values presented in brackets, because for the Age indicator, this is apparently SD, while for the other indicators, this is apparently %. However, this is not indicated anywhere.
P-values are calculated for each pair of indicators, so it is advisable to enter them along the appropriate line.
5. All generic names of bacteria, such as Klebsiella, Candida, Pseudomonas, etc. should be written in italics throughout the text and in tables.
After these corrections have been made, the manuscript can be recommended for publication.
Author Response
Response to Reviewer 3 Comments
Comment 1:
- 123–131: When describing the results, there is no clear understanding of the formation of the two groups of patients; therefore, the Materials and Methods section should include a schematic design for the patients who were included in the study and a rationale for those patients who were excluded from the study.
Response:
We thank the reviewer for this valuable suggestion. A schematic flowchart illustrating patient selection and exclusion criteria has been added to the results section to improve clarity.
Comment 2:
- 129: Clarify whether these patients were included at some point in the study, since these data are not presented in the tables.
Response:
We appreciate the reviewer’s attention to detail. We clarified in the results section that the 24 excluded patient records were identified during the initial screening process due to data quality issues and were not included in any of the statistical analyses or tables.
Comment 3:
Provide reference to all tables at the beginning of their description.
Response:
Thank you for the comment. We have revised the manuscript to ensure that each table is explicitly referenced at the beginning of its description to enhance readability and clarity.
Comment 4:
Table 1: Provide a designation for the values presented in brackets, because for the Age indicator, this is apparently SD, while for the other indicators, this is apparently %. However, this is not indicated anywhere. P-values are calculated for each pair of indicators, so it is advisable to enter them along the appropriate line.
Response:
We have added a footnote in Table 1 to clarify that continuous variables are presented as mean ± SD, and categorical variables are shown as number (percentage). Regarding the p-values, we clarified that a single p-value was computed per variable using chi-square tests for 2x2 comparisons, hence individual values are not applicable to every line.
Comment 5:
All generic names of bacteria, such as Klebsiella, Candida, Pseudomonas, etc. should be written in italics throughout the text and in tables.
Response:
We thank the reviewer for pointing this out. All genus names of microorganisms mentioned in the manuscript and tables have been revised to be written in italics, in accordance with scientific writing standards.